# In the Era of Systemic Therapy for Hepatocellular Carcinoma Is Transarterial Chemoembolization Still a Card to Play?

**DOI:** 10.3390/cancers13205129

**Published:** 2021-10-13

**Authors:** Ana-Maria Bucalau, Illario Tancredi, Gontran Verset

**Affiliations:** 1Department of Gastroenterology, Hepatopancreatology and Digestive Oncology, Hôpital Erasme, Université Libre de Bruxelles (ULB), 1070 Brussels, Belgium; gontran.verset@erasme.ulb.ac.be; 2Department of Interventional Radiology, Hôpital Erasme, 1070 Brussels, Belgium; Illario.Tancredi@erasme.ulb.ac.be

**Keywords:** hepatocellular carcinoma, transarterial chemoembolization, drug-eluting micro-spheres, immunotherapy, cone-beam computed tomography

## Abstract

**Simple Summary:**

Hepatocellular carcinoma (HCC) is a growing healthcare problem, with most of the cases occurring in patients with an underlying chronic liver disease. Transarterial chemoembolization (TACE) is recommended for unresectable tumors, mostly in a palliative setting. Several developments have seen the day during the last few years, with technique improvements in terms of efficacy and safety due to more selective therapies and better patient selection. Nevertheless, this is the era of systemic treatment for HCC, where immunotherapy and combination systemic treatments are taking the lead. As such, we have to ask ourselves, where does TACE stand today and is there a tomorrow?

**Abstract:**

Conventional transarterial embolization (cTACE) has been proven to be effective for intermediate stage hepatocellular carcinoma (HCC), with a recent systematic review showing an overall survival (OS) of 19.4 months. Nevertheless, due to the rapid development of the systemic therapeutic landscape, the place of TACE is becoming questionable. Is there still a niche for TACE in the era of immunotherapy and combination treatments such as atezolizumab–bevacizumab, which has shown an OS of 19.2 months with excellent tolerance? The development of drug-eluting microspheres (DEMs) has led to the standardization of the technique, and along with adequate selection, it showed an OS of 48 months in a retrospective study. In order to increase treatment selectivity, new catheters have also been added to the TACE arsenal as well as the use of cone-beam CT (CBCT), which provides three-dimensional volumetric images and guidance during procedures. Moreover, the TACE indications have also widened. It may serve as a “bridging therapy” for liver transplantation candidates while they are on the waiting list, and it represents a valuable downstaging tool to transplantation criteria. The aim of this review is to explore the current data on the advancements of TACE and its future place amongst the growing panel of treatments.

## 1. Introduction

Liver cancer is a growing health problem and currently represents the sixth most common cancer and fourth most frequent cause of cancer-related death worldwide, with hepatocellular carcinoma (HCC) accounting for the majority of cases [1]. Most cases of HCC are associated with cirrhosis, thus choosing the most suitable treatment option depends not only on the tumor stage but also on the severity of the underlying liver disease. The majority of current guidelines [2,3] consider the Barcelona Clinic Liver Cancer (BCLC) staging system as the algorithm of choice for tumor staging and therapeutic options, taking into account tumor burden, liver function, and performance status [4]. Few patients are diagnosed at an early stage, which is when a curative treatment is feasible, and the majority present intermediate or advanced HCC, which is suitable for palliative therapies.

Transarterial chemoembolization (TACE) is currently recommended as a first line therapy for intermediate HCC (BCLC B), more precisely, it is recommended for unresectable tumors with no vascular invasion or extrahepatic spread [2,3]. TACE involves the injection of a chemotherapeutic agent mixed with an embolic material and is administered as selectively as possible into the feeding arteries of the tumor, resulting in tumoral necrosis due to ischemia and cytotoxic effects [5]. It takes advantage of the double vascularization of the liver, with the tumor nodule’s blood supply being almost exclusively provided by the hepatic artery, while the majority of the parenchyma’s blood supply comes from the portal vein [6].Its benefits in terms of survival were described by two randomized trials (RCTs) [7,8] and are supported by a meta-analysis comprising 14 RCTs who showed a significant improvement in survival at 2 years compared with the control group [9]. 

The first technique described in the literature was conventional chemoembolization (cTACE), which consists of an intra-arterial injection of a lipiodol-chemotherapy suspension followed by embolization with Gelfoam particles. The development of drug-eluting microspheres (DEM) represented a step towards improvement in terms of the tolerance and safety profile, which is mainly due to the fact that the cytotoxic agent is released slowly, resulting in a reduced systemic passage [10,11,12]. Moreover, the arrival of new microspheres that are smaller and that have new properties as well as new catheters and the use of cone-beam computed tomography (CBCT) result in a more selective approach, increasing the safety and efficacy of TACE. With widened indications such as “bridging therapy” for liver transplantation candidates while on the waiting list or who are downstaging treatment to transplantation criteria and new associations with immunotherapy, this “old” technique is being dusted and repolished for future use.

In this review, we are going to take a look back on the path taken by chemoembolization, explore the current data on technical advancements, and try to find its future place among the growing panel of treatments.

## 2. Indications of TACE

According to the BCLC staging system, TACE is a first line therapy for BCLC B patients [4]. Chemoembolization became a standard of care after two RCTs showed a clear advantage in terms of survival compared to best supportive care (BSC). Llovet et al. reported a mean survival of 28.6 months after cTACE compared to 17.9 months in the case of BSC (*p* = 0.009) [7], while the team of *Lo* et al. depicted a clear superiority of cTACE vs. BSC at 1 year (57% vs. 32%), 2 years (31% vs. 11%), and 3 years (26% vs. 3%, *p* = 0.002) [8]. These results were confirmed by a meta-analysis by the same group, Llovet et al., who reported a higher 2-year survival with the TACE (odds ratio, 0.53; 95% confidence interval (CI), 0.32–0.89; *p* = 0.017) [9].

Ever since then, the indications of TACE have evolved and extended. Currently, it serves extensively as “bridging therapy” for liver transplant waiting list candidates [13].

Patients that have been listed may experience long waiting times and, consecutively, disease progression that is outside of transplantation criteria. This leads to an increased risk for dropout from the waiting list. Waiting times vary depending on the region, with dropout rates of patients awaiting liver transplantation ranging between 25% at 6 months, 38% at 12 months, and up to 55.1% at 18 months [3]. Therefore, bridging therapy has been suggested for all patients with HCC within transplantation criteria, with wait times >6 months [14]. As a bridge therapy, TACE has been shown to be effective by several studies, with lower dropout rates (3–13%) than previously reported [15]. Graziadei et al. showed no progression following TACE, and the five-year survival rates after liver transplantation were high at 93%, despite long waiting times prior to liver transplantation (mean of 178 days) [16]. Tumor recurrence was equally notably low at 2%.

Furthermore, TACE can represent a valuable downstaging tool to fulfill the transplantation criteria for patients presenting larger tumors, with success rates ranging from 24% [17] to 90% [18]. Moreover, a later study by Ravaioli et al. showed that at a median follow-up of 2.5 years after transplantation, the 1- and 3-year disease-free survival rates were comparable: 80% and 71% in the Milan group versus 78% and 71% in the downstaging group. In a recent prospective study with 200 patients that compared TACE used as “bridging therapy” or for downstaging, Affonso et al. did not find a difference in the 5-year post-transplant overall survival between the two groups: 73.5% in the downstaging group and 72.3% in bridging the group (*p* = 0.31. Recurrence-free survival was 62.1% in downstaging and 74.8% bridging groups (*p* = 0.93)) [19].

When contraindications to ablation, resection, or hepatic transplantation exist, the stage-migration concept advocates that the next best line of therapy, in this case TACE, should be applied for early stage HCC [2,3,20].

Finally, despite current guidelines not recommending TACE for advanced, BCLC C, disease, in the BRIDGE study, an international large scale and longitudinal cohort study that included 18031 patients from 14 countries, the authors found chemoembolization to be the first line treatment for more than 50% of the patients in this category [21]. Regarding portal vein tumor thrombosis (PVTT), in a prospective non-randomized study by *Luo* et al, cTACE significantly improved survival compared to BSC for both patients with PVTT of the segmental branches (*p* = 0.002) and patients presenting PVTT in the first order branches or in the main trunk of the portal vein (*p* = 0.002) [22]. Nevertheless, the survival was poor for all patients, with an overall median survival of 5.2 months.

## 3. Technical Aspects and Chemotherapeutic Agents

### 3.1. It All Started with Conventional Chemoembolization (cTACE)

TACE was introduced in Japan in the early 1980s and was designed to take advantage of the dual blood supply of the liver [23,24]. As described above, cTACE consists of an intra-arterial injection of a lipiodol-chemotherapy suspension followed by embolization with Gelfoam particles. The benefit of adding a chemotherapeutic agent versus bland embolization remains controversial, with data from the literature being scarce and with patient selection being heterogeneous. Yet, chemoembolization remains the standard of care for intermediate stage disease due to its proven efficacy in terms of survival compared to best supportive care [7,8]. 

In a systematic review, Lencioni et al. [25] showed that cTACE can reach a median overall survival (OS) of 19.4 months (95% confidence interval [CI]: 16.2–22.6) and a 5-year OS of 32.4% with an objective response rate (OR) of 52.5% (95% CI: 43.6–61.5). The incidence of post-embolic syndrome (PES) was 47.7%, and liver enzyme abnormalities were noted in 52% of patients. A low mortality rate was noticed, which reached 0.6%; the most common cause of death was liver insufficiency leading to hepatic failure. The authors emphasize the importance of patient selection in order to diminish complications post chemoembolization.

### 3.2. Heading toward a Standardization of TACE—Development of Drug Eluting Microspheres (DEMs)

In order to improve the efficacy and safety of chemoembolization and to standardize the technique, DEM-TACE was developed more than a decade ago. It represented a major improvement in terms of tolerance and safety profiles, mainly due to the fact that the cytotoxic agent is released slowly, resulting in a reduced systemic passage [10,11,12]. 

DEM-TACE involves the infusion of non-resorbable embolic microspheres loaded with a chemotherapeutic agent in order to achieve sustained drug release. Its “double purpose” is to deposit a high concentration of chemotherapy that is in contact with the tumor cell with a stable release and to induce tumor cell hypoxia by obstructing feeding arteries at the same time. Thus, microspheres are both drug-carriers and embolizing agents. This leads to a decrease in the systemic release of the drug due to the high-affinity carrier activity of the spheres and the absence of a time interval between injection and embolization since it does not require a second separate step, as is the case with cTACE.

Several types of microspheres are currently commercialized, with different sizes, ranging from 40 to 900 μm, and thus different chemotherapeutic agent loading and release times, as shown in two in vitro analyses [26,27] (Table 1). The loading time varies from 30 min to 2 h, and it increases with the size of the microspheres.

The first clinical study demonstrating the efficacy of DEM-TACE was a phase II study published by Varela et al. [28]. Highly selective DEM-TACE was performed using 500–700 μm microspheres on 27 Child–Pugh A cirrhotic patients, all of whom were stage BCLC B and with a mean tumor size of 46 mm (8–150). The objective response rate was of 66% (CR of 26%) after two procedures performed two months apart. The chemotherapeutic agent loaded on the microspheres was Doxorubicin, and its maximal peripheral concertation (Cmax) and area under the curve AUC were significantly lower than they were in conventional TACE (*p* = 0.00002 and *p* = 0.001, respectively). After a median follow-up of 27.6 months, the 1- and 2-year survivals were 92.5% and 88.9%, respectively. Severe adverse events (SAE) were only noted in two patients in the form of liver abscesses, with one leading to the death of the patient. 

The key data evaluating DEM-TACE came from two retrospective studies from Burrel et al. and Malagari et al. [29,30]. They both emphasized the importance of patient selection and showed a median survival of 48.6 months and 43.8 months, respectively, in selected patients with preserved liver function and who had been diagnosed with BCLC A and B disease. In the subgroup analysis, patients presenting an intermediate stage disease had a lower median survival than those in the early stage group: 47.7 months vs. 54.2 months [29]. Regarding liver function, cumulative survival was also better for the Child–Pugh A group compared to the Child–Pugh B group (*p* = 0.029), with superior results for smaller tumors (<5 cm) translating into higher 5-year survival rates of 47.6% vs. 23.5%, respectively [30]. In the latter study, the PES incidence was 73.9% and was treated symptomatically across all sessions in 173 patients with HCC who underwent chemoembolization with 100–300 μm and/or 300–500 μm DEMs.

The most used microspheres are DC Beads (Boston International, London, UK), which are hydrophilic, nonresorbable, and precisely calibrated hydrogel microspheres that are biocompatible. They are available in four different ranges: 100–300, 300–500, 500–700 μm, and, more recently, there are also DC Beads M1 70–150 μm that have been evaluated in recent prospective and retrospective studies that presented an OR of 77 to 93% [31,32,33]. Liver and biliary injuries occurred in 32% of patients, all of whom were asymptomatic. Furthermore, DC Beads were the first to also be developed into a radiopaque version, DC Beads LUMI^TM^, which are designed to be visible under imaging (computed tomography (CT), cone-beam computed tomography (CBCT), and Fluoroscopy). These biocompatible, nonresorbable hydrogel spheres, which are produced from polyvinyl alcohol-like conventional DC Beads and that range from 70–150 μm, show a similar pharmacokinetic profile to conventional DC Beads and aim to see the spheres themselves in real-time, thus providing intraprocedural feedback on treated areas and allowing for a better evaluation of the tumor coverage. *Aliberti* et al. evaluated the feasibility and safety of LUMI Beads in 44 HCC patients [34]. TACE had no intraprocedural complications. The observed side effects were of mild intensity and included pain in five (11%), fever in four (9%), and vomiting in two (5%) patients. Most patients (89%) reported no adverse events. In the USA, LC Beads LUMI^TM^ are commercialized in two ranges: 70–150 and 40–90 μm. In a retrospective multi-center study that included 82 patients *Lakhoo* et al. showed an OR and disease control rate (DCR) of 47.6% (CR 19.5%) and 76.8%, respectively [35]. Grade 3 adverse events were seen in 6.1% of patients.

HepaSpheres (Merit Medical, MA, USA), which are also biocompatible, nonresorbable, and expandable, are another type of microspheres and have a diameter that ranges between 30–200 μm and are made from two monomers (vinyl acetate and methyl acrylate) that combine to form a copolymer (sodium acrylate alcohol copolymer). In 2015, *Malagari* et al. reviewed the available data on local response in patients with intermediate and early stage hepatocellular carcinoma that was nonresponsive to curative treatments and showed complete response and partial response rates ranging from 22.2 to 48% and 43.7 to 51%, respectively [36]. Mild PES was reported in 18–85.9% of patients. A year earlier, the same team had shown a survival rate of 100% at 12 months in a prospective study [37].

The sphere panel also comprises Embozene TANDEM microspheres (CeloNova Biosciences/Boston Scientific, MA, USA), a nonresorbable poly-metacrylate hydrogel that ranges in size from 40 to 100 μm; CalliSpheres® (Jinyuan Pharmaceutical Manufacturing Co., Ltd., Changzhou, China), the first microspheres developed in China and that are made from polyvinyl alcohol hydrogel and are available in five different sizes, ranging from 100 μm to 1200 μm; and LifePearls ® (Terumo NV, Leuven, Belgium), which are polyethylene glycol DEMs, 100–400 μm.

While Callispheres are available on in Asia, Embozene Tandem and LifePearls ® can be used in Western countries.

The MIRACLE I pilot study showed that 75 μm TANDEM spheres achieved tumor response or stable disease in 95% of patients, with a one-year survival rate of 56% [38].

*Aliberti* et al. treated 42 HCC patients with 100 µm Doxorubicin-charged LifePearls ® and obtained an overall tumor response rate of 79% (50% CR) at 1 month [39]. Fever (33%), increase in transaminase levels (17%), and pain (33%) were the most frequent adverse events, and their intensity was mostly mild (grades 1 and 2). In a single-center retrospective study with 302 patients, response was obtained in 85.5% of patients (63.2% CR), with a low occurrence of AEs (one liver abscess) [40]. Survival analysis at 12 months showed a progression-free survival (PFS) rate of 65.9% and an OS rate of 93.5%. Moreover, a recent review on the data from LifePearls ® trials shows an OR ranging from 79% to 88.5% [41]. The recently published PARIS Registry showed OR and DCR of 81% and 99%, respectively, as the best responses. The survival rates at one and two years were 81% and 66%, respectively, while the median OS was not reached. The median PFS was 13.7 months (95% CI: 11.3; 15.6), and the median time to TACE untreatable progression was 16.7 months (95% CI: 12.7; not estimable (n.e.)) [42]. However, the most interesting point of this study was the 15.5% rate of hepatobiliary toxicities (HBT), which is lower than the rate that was previously described in the literature. *Monier* et al. reported HBT after 14.4% of procedures with cTACE and 36.8% with DEM-TACE [43], while *Guiu* et al. reported HBT after 4.2% for cTACE and 30.4% for DEM-TACE in a population of cirrhotic patients, with a lower incidence of bile duct injury reported in cirrhotic versus non-cirrhotic patients [44].

As displayed above, a large panel of microspheres is available; nevertheless, in the absence of comparative trials, the choice of using DEMs remains in the hands of physicians. However, several studies advocate the use of smaller spheres, as they have been linked to higher drug dose administration and improved tumor penetration and tumor coverage. 

Padia et al. reported a study of 61 patients with HCC treated with either 100–300 µm or 300–500 µm microspheres loaded with 50 mg of doxorubicin. They found a significantly lower incidence of PES after treatment in the 100–300 μm group (36%) versus the 300–500 μm group (70%). The mean change in tumor size was similar between the two groups; nevertheless, there was a trend toward a higher incidence of CR with the 100–300 versus 300–500 μm beads (59 vs. 36%; *p* = 0.114) [45].

This led to a comparison between the 70–150 and 100–300 µm beads by *Lewis* et al., who indicated that smaller (70–150 μm) beads should be able to permit an increased dose and a more distal penetration of the drug to be administered to both hypervascular and hypovascular tumors compared to 100–300 µm beads in order to improve outcome [46].

Moreover, Prajapati et al. showed that TACE with 100–300 µm-sized DEMs is associated with a significantly higher survival rate (15.1 and 11.1 months, respectively (*p* = 0.005)) and lower complications than TACE with 300–500 and 500–700 µm-sized DEMs [47].

Although the choice of microsphere is not based on standardized recommendations, these findings underline the efficacy and safety of DEM-TACE, with a trend towards smaller particles for safer and more efficient treatment. 

### 3.3. Predictive Factors of Response

Several predictive response factors to TACE have been described, such as baseline tumor burden, tumor biology in terms of tolerance to ischemic stress and sensitivity to chemotherapeutic agents, alpha-feto protein(AFP), and liver function factors (e.g., albumin and bilirubin) as well as procedure-related factors [48,49]. 

Moreover, several TACE sessions might be necessary in order to improve response and survival [50,51]. In a retrospective study, *Kim* et al. showed that both the initial (adjusted hazard ratio (HR) 0.410) and the best response (adjusted HR 0.335) predict OS effectively [52]. Patients with complete response (CR) according to the modified Response Evaluation Criteria in Solid Tumors (mRECIST) [53] as the initial response had the longest OS followed by those who subsequently achieved CR after at least two sessions and those who achieved partial response (PR) as the best response (70.2, 40.6, and 23.0 months, respectively; log-rank test, *p* < 0.001). Large (>5cm) and multiple (>3) tumors were independently associated with failure to achieve CR after initial TACE (both *p* < 0.05). 

Furthermore, it is well known that angiogenic factors have been associated with poor prognosis, for example, serum vascular endothelial growth factor (VEGF) or hypoxia inducible factor-1 alpha (HIF-1), which tend to increase more in suboptimal responders compared to those with CR after TACE [54,55,56]. 

Since several TACE sessions might be required to achieve response, one important question arises: when to continue or to stop treatment. This question is even more relevant in the present setting, where a large panel of systemic treatments is available for patients with a preserved liver function, giving us a limited window of opportunity and choice. Several scores have been developed in order to improve patient selection either for the first session of TACE (selection for transarterial chemoembolization treatment (STATE) and hepatoma arterial-embolization prognostic (HAP)) or to evaluate the benefits of continuing therapy (Assessment for Retreatment with TACE (ART) and alfa-protein, Barcelona Clinic Liver Cancer, Child–Pugh, and response (ABCR)) [49,57,58,59]. Each score uses different liver function parameters, tumor burden or response, to assess the interest of treatment or retreatment. Nevertheless, their predictive value has not yet been shown, and they are just a step in the decision-making process. In current practice, TACE should be discontinued if local or extrahepatic progression appears or if there is a deterioration of the liver function or performance status or if no radiological response is attained after two sessions [60]. 

In order to better understand how and when to use TACE, in a recent review, *Raoul* et al. proposed an algorithm based on expert opinions and clinical evidence [61]. 

### 3.4. Type of Chemotherapeutic Agents

The use of a chemotherapeutic agent associated with embolization is still under debate. A randomized trial conducted from 2007–2012 compared the outcome of embolization using microspheres alone (TAE) with chemoembolization using doxorubicin-eluting microspheres. There was no apparent difference between the treatment arms for TAE and TACE, with no difference in response: 5.9% versus 6.0%, respectively (difference, −0.1%; 95% CI, −9% to 9%). The median PFS was 6.2 versus 2.8 months (HR, 1.36; 95% CI, 0.91 to 2.05; *p* = 0.11), and the overall survival was 19.6 versus 20.8 months (HR, 1.11; 95% CI, 0.71 to 1.76; *p* = 0.64) for TAE and TACE, respectively [62].

**Table 1 cancers-13-05129-t001:** Results in terms of efficacy of major studies on conventional and drug-eluting transarterial chemoembolization of microspheres.

Authors	Year	Technique	Diameter of Particles	Drug	Number of Patients	Objective Response	Survival Rate	Ref.
Llovet et al.	2002	cTACE vs. BSC vs. TAE	NA	Doxorubicin	112	-	82% (1y) 63% (2y)	[7]
Lo et al.	2002	cTACE VS BSC	NA	Cisplatin	80	39%	57% (1y) 31% (2y) 26% (3y)	[8]
Varela et al.	2007	DEM-TACE	500–700 µm	Doxorubicin	27	75%	92.5% (1y) 88.9% (2y)	[28]
Burrel et al.	2012	DEM-TACE	300–500 µm 500–700 µm	Doxorubicin	104	-	89.9% (1y) 38.3% (5y)	[29]
Malagari et al.	2012	DEM-TACE	100–300 µm 300–500 µm	Doxorubicin	173	35,2% (after TACE1)	93.6% (1y) 22.5% (5y)	[30]
Spreafico et al.	2014	DEM-TACE	70–150 µm	Doxorubicin	45	77.7% (1mo)	-	[31]
Deipolyi et al.	2014	DEM-TACE	100–300 µm vs. 70–150 µm+ 100–300 µm	Doxorubicin	84	21% vs. 24% (1mo)	-	[33]
Malagari et al.	2014	DEM-TACE	30–60 µm	Doxorubicin	45	68.9%	100% (1y)	[37]
Richter et al.	2017	DEM-TACE	75 µm	Doxorubicin	25	67% (1mo)	56% (1y)	[38]
Aliberti et al.	2017	DEM-TACE	100 µm	Doxorubicin	42	79% (1mo)	-	[39]
Veloso et al.	2018	DEM-TACE	100–200 µm	Doxorubicin	302	85.5 (1mo)	93.5%	[40]
Guiu et al.	2019	DEM-TACE	100–300 µm	Idarubicin	46	68% (6mo)	63% (1y)	[62]
Lakhoo et al.	2020	DEM-TACE	70–150 µm (radiopaque)	Doxorubicin	82	56%	94.6% (1y)	[34]
De Baere et al.	2020	DEM-TACE	100–200 µm	Doxorubicin Idarubicin	97	81%	81% (1y) 66% (2y)	[42]

cTACE: conventional transarterial chemoembolization; BSC: best supportive care; TAE: transarterial embolization; DEM-TACE: drug-eluting microspheres transarterial chemoembolization.

Nevertheless, in an animal liver tumor model, 20 rabbits were treated transarterially with doxorubicin-loaded 70–150μm DEMs. The study showed that incremental increases in doxorubicin correlates with greater necrosis in rabbit liver tumors after DEM-TACE that supports the use of chemotherapeutic drugs in transarterial therapy [63].

A systematic review of chemotherapeutic regimens in 52 studies of TACE showed that the most widely used anticancer drugs were doxorubicin (36%), cisplatin (31%), epirubicin (12%), mitoxantrone (8%), and mitomycinC (8%) [64], with no indication toward the superiority of one of the drugs in particular. An in vitro study conducted by Boulin et al. showed that the drug that was the most effective on three HCC cell lines was another anthracycline, idarubicin [65]. Nevertheless, the most used chemotherapeutic agent in western countries is doxorubicin, and more precisely, this was the drug that was used in the majority of the studies mentioned above. In a recent phase II study, IDASPHERE II, *Guiu* et al. evaluated the 6-month ORR after DEM-TACE with idarubicin [66]. Forty-four patients received treatment with a 6-month OR of 52%. The median PFS, time to progression (TTP), and OS were 6.6 months, 9.5 months, and 18.6 months, respectively. TACE was discontinued for toxicity in 9% of participants and the most frequent grade 3–4 adverse events were mainly biological abnormalities and pain (16%). 

It is less likely that further studies will be conducted to compare TACE and TAE; nevertheless, the quest for the most efficient chemotherapeutic agent will probably continue with the development of idarubicin-loaded microspheres.

### 3.5. cTACE Versus DEM-TACE: Frenemies

Debate still exists as to when to choose between these two techniques, with current literature still not being able to clarify the superiority of DEM-TACE over cTACE; nevertheless, there are some benefits in terms of safety when choosing embolizing microspheres. However, the refinement in patient selection and more standardized selective treatment targeting a personalized treatment remain the key element for the improvement on intra-arterial therapies.

The PRECISION V phase II study was a multicenter RCT published in 2010 that included 212 patients with Child–Pugh A/B cirrhosis and large and/or multinodular, unresectable tumors. Although it failed to show superiority in terms of tumor response at 6 months for DEM-TACE compared to cTACE (*p* = 0.11), it demonstrated that patients with Child–Pugh B, ECOG 1, bilobar disease, and recurrent disease had a significant increase in objective response (*p* = 0.038) when treated by DEM-TACE compared to cTACE. Moreover, the use of microspheres loaded with doxorubicin was associated with improved tolerability, with a significant reduction in serious liver toxicity (*p* < 0.001) and a significantly lower rate of doxorubicin-related side effects despite a higher mean dose being administrated (*p* = 0.0001) [11]. *Golfieri* et al. rechallenged the question in 2014, demonstrating similar results between the two techniques in terms of tumor response and TTP at 9 months. Furthermore, no difference was seen in terms of OS at 1 and 2 years, with 83,5% and 55.4% after cTACE and 86.2 and 56.8% after DEM- TACE (*p* = 0.949), respectively. Furthermore, the study was terminated prematurely for futility. In terms of safety, no significant difference in the incidence of all AEs was observed, with the exception of post-procedural pain, which was lower when using microspheres (Grade 3: 18.2% for cTACE versus 0% for DEM-TACE) [12].

Over the years, several studies have addressed the issue of cTACE versus DEM-TACE with contradictory results (Table 2). 

In the early 2010s, three studies found that therapy with DEMs offered a survival advantage over conventional chemoembolization for patients with unresectable HCC. Dhanasekaran et al. disclosed a significantly longer median survival with DEM-TACE vs. cTACE: 610 (351–868) and 284 days (4–563; *p* = 0.03), respectively [67]. Their results were consistent with those reported by Wiggermann et al, where a longer mean survival was shown when using microspheres: 651 ± 76 days versus 414 ± 43 days for cTACE (*p* = 0.01) [68]. In the same perspective, Song et al. demonstrated a significant difference in the tumor response rate (CR: 55 vs. 23.1% and OR: 81.6 vs. 49.4%) and in the OS (32.2 ±1.9 months vs. 24.7± 1.7 months; *p* = 0.005) when using DEB-TACE over cTACE [69].

Opposite to these findings, in 2016, Facciorusso et al. showed an advantage of using cTACE over DEM-TACE in terms of OR (85.3% vs. 74.8%, *p* = 0.039) and PFS (median PFS 17 months versus 11 months, *p*< 0.001). No difference was seen in survival (39 months vs. 32 months, *p* = 0.10); nevertheless, there was a favorable trend in the conventional group for patients with bilobar neoplasia, portal hypertension, and elevated AFP [70]. *Lee* et al. addressed the issue in a retrospective study that stratified patients by index tumor size. No significant difference was seen in the OS between the two groups (46.6 months for DEM-TACE and 44.9 months for cTACE, respectively; *p* = 0.660) nor in the disease control rate at 1 month (78.3% vs. 86.8%; *p* = 0.076) despite of tumor size [71]. However, in terms of safety, patients with tumors exceeding 5 cm presented higher complication rates when treated with cTACE rather than DEM-TACE (14.6 vs. 6.6%; *p* = 0.04). 

More recently, the Scandinavian team of Karalli et al. retrospectively analyzed data from 202 patients and found no difference in survival between the two therapies, with a median OS of 17.1 months in the cTACE group and 19.1 months in the DEB-TACE (NS) group. Nevertheless, DEM-TACE had better tolerability compared to cTACE, with patients presenting less abdominal pain (48% vs. 64%, *p* < 0.05), nausea and vomiting (36% vs. 51%, *p* < 0.05), fever (28% vs. 43%, *p* < 0.05), or fatigue (20% vs. 33%, *p* < 0.05) [72].

A five-year follow-up comparison published by *Liu* et al. showed that a greater percentage of patients treated with cTACE died than those treated with DEB-TACE (76.1% vs. 66.7%) (*p* = 0.045). Nevertheless, the median survival time was 37 months in both treatment groups. The median time to disease progression was in favor of DEM-TACE: 16 months vs. 11 months (*p*  =  0.019) [73]. 

Very recent data from Gjoreski et al. also showed a comparable survival between the two techniques. The overall 12- and 24-month survival rates were 85.7 and 63.6% after c-TACE and 90.2 and 75.8% after DEM-TACE (*p* = 0.18). No significant difference in terms of adverse events was found. Nevertheless, DEM-TACE requires a shorter in-hospital stay [74].

When used for the treatment of early disease in the stage-migration setting, a retrospective study including 76 patients showed no difference in the 1-year OR rates, which were 85% and 88.9% for cTACE and DEM-TACE, respectively (*p* = 0.935), nor a survival benefit (*p* = 0.603) [75].

What is interesting to add is that the debate continues in the different meta-analyses or systematic reviews that have been published so far. 

*Martin* et al. conducted a systematic review and found that DEM-TACE had a significant advantage compared to cTACE in terms of OR and had greater overall DCR in patients with advanced HCC (*p* ≤ 0.038) [76]. Their findings were consistent with the meta-analysis by Zhou et al., which included nine studies and total of 830 patients [77]. They found DEM-TACE to significantly improve overall survival and increase OR and DCR. *Huang* et al. found a significantly better OR for DEM-TACE than cTACE (*p* =  0.004) and a significant improvement in the 1- and 2-year survival (*p* ≤0.007). However, they found no difference between treatments for the 6-month and 3-year survival rates (*p* ≥0.11) [78]. Another meta-analysis, by Zou et al. found that DEM-TACE was associated with a higher complete response (OR 1.35) and a higher OS rate (OR, 1.41) [79]. All four studies found that DEM-TACE was associated with fewer side effects compared to cTACE.

However, a more recent meta-analysis performed by Facciorusso et al. that included 12 studies with 1449 patients, four of which were randomized controlled trials, observed that the 1-year (odds ratio: 0.76, 0.48–1.21, *p* = 0.25), 2-year (odds ratio: 0.68, 0.42–1.12, *p* = 0.13), and 3-year survival (odds ratio: 0.57, 0.32–1.01, *p* = 0.06) did not differ between treatments [80]. No statistically significant difference in adverse events was registered (odds ratio: 0.85, 0.60–1.20, *p* = 0.36).

In a Chinese population treated with CalliSpheres, a meta-analysis of 16 studies with 1454 HCC patients by Liang et al. showed a higher 1-month and 3-month OR (odds ratio: 2.87, 95% CI: 2.15–3.83 and odds ratio: 3.39, 95%CI: 2.45–4.70) and disease control rate in favor of microsphere treatment (odds ratio: 2.01, 95% CI: 1.37–2.95 and odds ratio: 1.71, 95%CI: 1.14–2.55) [81]. No difference in the PFS, OS, adverse events, or liver function was observed between the two therapies.

No significant difference in survival was seen by *Han* et al. at 1-year (OR 1.51, 95% CI 0.48- 1.21, *p* = 0.08) and 2 years (odds ratio 1.32, 95% CI 0.74–2.36, *p* = 0.34) after TACE [82]. However, the 3-year survival rate was significantly higher in patients who underwent DEM-TACE (odds ratio = 1.92, 95% CI = 1.00–3.67, *p* = 0.049). The safety was similar between C-TACE and DEB-TACE.

If the question of which technique to use remains of interest and is still debated, then the importance of good patient selection and adequate indication remains essential for the good use of these intra-arterial treatments.

**Table 2 cancers-13-05129-t002:** Results in terms of efficacy of selected studies comparing conventional and drug-eluting microsphere transarterial chemoembolization.

Authors	Year	Technique	Diameter of Particles	Drug	Number of Patients	Objective Response	Survival Rate	Ref.
Lammer et al.	2010	cTACE vs. DEM-TACE	300–500 µm 500–700 µm	Doxorubicin	93	43.5% vs. 51.6% (6mo)	-	[11]
Wiggermann et al.	2011	cTACE vs. DEM-TACE	300–500 µm	Epirubicin	22	22.7% vs. 22.7% (8w)	55% vs. 70% (1y)	[68]
Song et al.	2012	cTACE vs. DEM-TACE	100–500 µm	Doxorubicin	129	26.6% vs. 55% (3mo)	80% vs. 88% (1y)	[69]
Dhanasekaran et al.	2013	cTACE vs. DEM-TACE	300–500 µm 500–700 µm	Doxorubicin	71	-	46% vs. 67% (1y) 19% vs. 40% (2y)	[67]
Golfieri et al.	2014	cTACE vs. DEM-TACE	100–300 µm	Doxorubicin	177	89.7% vs. 92.1% (1mo)	83.5% vs. 86.2%(1y) 55.4% vs. 56.8% (2y)	[12]
Facciorusso et al.	2016	cTACE vs. DEM-TACE	100–300 µm	Doxorubicin	246	85.3% vs. 74.8% (1mo)	35.3% vs. 43.9% (1y)	[70]
Lee et al.	2016	cTACE vs. DEM-TACE	Not mentioned	Doxorubicin	250	86.8% vs. 78.3%	>90% (1y)	[71]
Liu et al.	2018	cTACE vs. DEM-TACE	300–500 µm	Doxorubicin	273	-	38% vs. 23% (5y)	[73]
Karalli et al.	2020	cTACE vs. DEM-TACE	100–300 µm 300–500 µm	Doxorubicin	202	-	60–80% (1y)	[72]
Kang et al.	2020	cTACE vs. DEM-TACE	75–150 µm	Doxorubicin	76	82.5% vs. 94.4% (1mo)	50% vs. 47.3% (1y)	[75]
Gjoreski et al.	2021	cTACE vs. DEM-TACE	100-400 µm	Doxorubicin	60	-	85.7% vs. 89.8% (1y) 63.6% vs. 85.7%	[74]

cTACE: conventional transarterial chemoembolization; DEM-TACE: drug-eluting microspheres transarterial chemoembolization.

## 4. There Is Always Room for Improvement

Improving treatment by refining the TACE technique is one of the goals of the new tools that have recently been developed. In a review from 2016, *Miyayama* et al. has already shown the importance of supraselective chemoembolization, mainly for patients with Child–Pugh scores of 5–8, tumors larger than 7 cm, and more than five lesions [83].

### 4.1. More Precise with New Catheters

Balloon-occluded TACE (B-TACE) was first developed in Japan by *Irie* et al. in order to improve the results of cTACE [84]. This technique is performed using an occlusive balloon microcatheter inflated in the arterial feeders of the tumor nodules before embolization, thus inducing a drop in local blood pressure that allows a blood flow modification with a higher concentration of chemotherapy at the tumor level and sparing the nontumoral parenchyma. 

Results have been promising in several studies that have combined B-TACE with lipiodol-based chemoembolization [85,86,87], and retrospective data have suggested B-TACE has better tumor control compared to cTACE in tumors that are up to 4 cm in diameter [85,88,89]. 

The first two retrospective studies evaluating B-TACE using DEMs (B-DEM-TACE) in HCC came from Lucatelli et al. and Goldman et al. [90,91].

The first study used exclusively B-DEM-TACE, whilst the latter used combined conventional and microsphere balloon-occluded TACE.

In an Italian single-centre retrospective study, 22 patients were treated with epirubicin-loaded polyethylene-glycol (PEG) microspheres (100  ±  25 µm and 200  ±  50 µm) in order to evaluate the technical success, safety profile, and oncological results of balloon-occluded transcatheter arterial chemoembolization. Exclusive target embolization was achieved in 14/24 procedures (58.3%). AEs occurred in 17% of patients, with one grade 3 pseudo-aneurysm of the feeder. PES occurred in 33% of patients, which was lower than the incidence reported in the existing literature with TACE without balloon. The OR at 1 and 3–6 months was 90.9% and 58.3%, respectively. 

Goldman et al. showed comparable results in terms of OR that was superior to 90% (60% CR and 33.3% PR). The technical success rate was 100% (28 of 28 cases), with 1 minor complication of left portal vein thrombosis and small liver infarct.

In 2020, our team published the first prospective evaluation of B-TACE using PEG embolizing microspheres loaded with doxorubicin that included 24 patients with a 100% technical success rate [92]. Clinical grades 1/2 were reported in 25.7% of patients, with abdominal pain being the most frequent complication (17.1%), and radiological evaluation disclosed two cases of biloma/liver infarct that did not impair further treatment. The overall OR at 1-month post chemoembolization was of 74.3%. The key message from the study was the preservation of liver function in order to provide the opportunity to receive a sequential treatment to the patient in case of progression since the development of various systemic therapies that improve OS of these patients. All 24 patients were candidates for multimodal treatment after B-DEM-TACE due to the lack of hepatic injury.

The first case–control study comparing B-DEM-TACE vs. DEM-TACE came from the Italian team of Lucatelli et al. [93]. The objective response was similar between the two techniques at 1 and 3 months but was favorable for B-DEM-TACE at 9–12 months (78.9% vs. 53.9%, *p*  =  0.05). Moreover, the time to recurrence for the complete responders was also favorable for the use of balloon-occluded TACE (278.0 days [196.0–342.0] vs. 219.0 days [161.0–238.0], OR 0.68 [0.4–1.0], *p*  =  0.10). No significant differences were observed in terms of safety.

The latest data come from a retrospective multicentric study that included 96 patients and compared the tumor response rates of B-TACE to non-B-TACE (more precisely DEM-TACE or cTACE) using propensity score matching (PSM) [94]. Moreover, they evaluated the clinical benefit of this new catheter, translated by lower rates of TACE re-intervention achieved using B-TACE. The best target OR after PSM were similar between the two groups (90.1% vs. 86.8%, *p*  =  0.644); however, the CR at 1–6 months was significantly higher for B-TACE (59.3% vs. 41.8%, *p*  =  0.026), and the retreatment rate was lower for B-TACE (9.9% vs. 22.0%, *p*  =  0.041). In terms of safety, there was a very important difference in terms of the occurrence of PES, with 8.8% in non-B-TACE and 41.8% in B-TACE (*p*  <  0.001), all Grade 1–2. The authors explain this as being due to the drug infusion and absorption both in the tumor and in the peritumoral area, which could have contributed to the higher CR rates that were achieved.

The Surefire Infusion System (SIS) is an anti-reflux microcatheter which has a funnel-shaped, self-expanding tip that partially collapses during systolic flow but that expands during diastole, providing a barrier to prevent particle reflux. Literature regarding the use of this catheter is still poor. A retrospective single center evaluating tumor response after DEM-TACE delivered with SIS showed promising results, with disease response in 91% of patients and 85% of lesions after a single treatment [95]. Safety profiles were acceptable.

The SeQure® microcatheter (Accurate Medical Therapeutics, Guerbet, Paris, France) is a new reflux control microcatheter that produces a local fluid barrier based on flow dynamics in order to deliver more microspheres to targeted vessels and are associated with a decreased risk of non-target embolization. It consists of side slits that are specifically sized to allow the outflow of contrast media, creating a fluid barrier around the microcatheter that prevents the back passage of the embolization microspheres along the catheter, thus reducing their reflux. A very recent study compared the differences in non-target embolization and vessel microsphere filling of the SeQure® microcatheter compared to a standard microcatheter in a swine model [96]. It reduced the risk of non-target embolization from 11% to 1.7%, increasing the delivery of microspheres to 98% of the target vessels compared to standard microcatheters. 

### 4.2. More Selective with Cone-Beam Computed Tomography (CBCT)

It has been shown that technical factors such as selective embolization contribute to the survival of HCC patients [97]. C-arm cone-beam CT (CBCT) is a useful tool for obtaining cross-sectional and three-dimensional (3D) images during interventional procedures. In HCC, it can provide important additional diagnostic information, including visualization of small tumors and their feeding-arteries (hepatic or extrahepatic) and possibly of extrahepatic collateral arteries. A recent meta-analysis comprising 18 studies showed that CBCT can significantly increase the detection of tumors and tumor feeding arteries during TACE [98].

Moreover, it can be used for the detection of remnant viable tumor after TACE (Figure 1).

In a retrospective evaluation in 2011, Iwazawa et al. depicted that C-arm CT is nearly equivalent to biphasic multidetector computed tomography (MDCT) for detecting incomplete iodized oil accumulation after cTACE; more precisely, they determined that it helps to recognize a suboptimal treatment immediately post TACE [99]. Several studies followed, supporting these findings [100,101], among which was a recent retrospective analysis by Orlacchio et al. that assessed the ability of CBCT to predict short-term response at the 30-day follow-up CT after TACE [102]. Evaluation of the area under the ROC curve showed that the diameter, volume, and density of the lesion measured with CBTC had an accuracy of 94%, 96%, and 98%, respectively, in discriminating a complete response from a not complete response. As mentioned earlier, tumor response is the most important predictive factor of survival; this early assessment of treatment is key to future patient management. *Choi* et al. also found an advantage of using CBCT immediately after TACE for assessing response and for predicting the response outcome [103].

In a large retrospective study on 207 HCCs ≤ 6 cm, Miyayama et al. showed that in patients who were treated with supraselective TACE, intraprocedural CBCT reduces local recurrence compared to the standard digital subtraction angiography (DSA), with 1-, 2-, and 3-year local recurrence rates in the DSA vs. CBCT of 33.3% and 22.3%, 41.3% and 26.8%, and 48% and 30.6%, respectively (*p* = 0.0217) [104].

Furthermore, Iwazawa et al. reported that the OS rates of patients who underwent chemoembolization with and without C-arm CT assistance were 94% and 79%, 81% and 65%, and 71% and 44% at 1, 2, and 3 years, respectively [105]. The local progression-free survival rates of these patients were 43% and 27%, 31% and 10%, and 26% and 5% at 1, 2, and 3 years, respectively. Multivariate analysis showed that C-arm CT assistance was an independent factor associated with longer overall survival (hazard ratio, 0.40; *p* = 0.033) and local progression-free survival (hazard ratio, 0.25; *p* = 0.003).

## 5. Combined Treatments with TACE: Past, Present and Future

### 5.1. TACE and Local Ablation

According to current guidelines, surgery or liver transplantation are the indicated treatment for BCLC 0 (very early) or A (early disease) [2,3]. Nevertheless, in case of contraindications to these surgeries, percutaneous ablation, such as radiofrequency or microwave ablation, are indicated. The basis of combination treatment between these locoregional treatments were set in 2008 by Mostafa et al. [106]. They looked to determine the optimum combination strategy in an experimentally induced hepatic tumor model. Better results in terms of coagulation areas were obtained when TACE was performed before RFA rather than RFA before TACE or when RF ablation or TACE were performed alone. Furthermore, better histopathological results were obtained when TACE rather than bland embolization was performed before RFA, underlining the importance and synergy of the chemotherapeutic regimen.

In 2010 in a meta-analysis of RCTs with a total of 595 patients, Wang et al. reported that combining TACE with percutaneous ablation (RFA or percutaneous ethanol injection, PEI) improved the 1-, 2-, and 3-year OS compared to that of monotherapy (odds ratio 2.28, 95% CI 1.14–4.57; *p* = 0.020; odds ratio = 4.53, 95% CI 2.62–7.82, *p* < 0.00001 and odds ratio = 3.50, 95% CI 1.75–7.02, *p* = 0.0004, respectively) [107]. Sensitivity analysis demonstrated no difference in survival for TACE plus RFA vs. RFA for patients with small HCCs, probably because RFA already achieves complete necrosis in 90% of small (<3 cm) nodules, making a combined treatment redundant. 

Two RCTs from the team of Peng et al. showed a clear advantage in survival for combination therapy TACE plus RFA for larger nodules [108,109]. First, in a cohort in 139 patients with recurrent HCC up to 5 cm in diameter in 2012, they showed that the 1-, 3-, and 5-year overall survival rates were in favor of combination treatment 94%, 69%, and 46% versus 82%, 47%, and 36% for the RFA group (*p* = 0.037). In 2013, in a cohort of 189 patient with larger solitary tumors that were up to 7 cm or that had a maximum of three lesions that measured less than 3 cm, they also reported a significantly better OS for cTACE plus RFA versus RFA alone (HR, 0.525; 95% CI, 0.335 to 0.822; *p* = 0.002; HR, 0.575; 95% CI, 0.374 to 0.897; *p* = 0.009, respectively).

In patients with tumors rising up to 10 cm, the 1-, 3-, 5-year OS rates were 48.1% vs. 76.2%, 6.5% vs. 37.1%, and 0 vs. 16.4% between the cTACE group and the cTACE plus RFA group (RFA performed usually 7–15 days after TACE). The median OS was 12.00 months (8.88–15.13 months) in the TACE group and 27.57 months (20.06–35.08 months) in the TACE + RFA group (*p* < 0.001) [110]. 

The same favorable results were reported for recurrent HCC after hepatectomy in a recent propensity score matching study, with a better 5-year OS (41.6% vs. 30.2%, *p* = 0.028) and 5-year PFS rate (21.3% vs. 15.8%, *p* = 0.024) being observed for TACE plus ablation (median of 26 days, range 2–46 days after chemoembolization); in this case, RFA or microwave ablation (MWA) OS was better than that of TACE alone [111]. 

When compared to surgery, a recent meta-analysis of eight retrospective studies and one RCT reported no significant difference in the 1-year, 3-year and 5-year OS or DFS between TACE plus RFA and surgery [112]. TACE plus RFA had a higher local tumor progression rate (odds ratio 2.48, 95% CI 1.05–5.86, *p* = 0.04) compared to surgery, but the intrahepatic distant recurrence and distant metastasis rates were not significant. This suggests that for a specific population of patients not fit for surgical treatment, TACE plus RFA might be an equivalent option. However, this remains a debatable subject with data that also points to the superiority of surgical treatment [113,114].

When asking what percutaneous ablation therapy to combine with TACE, a recent meta-analysis showed a benefit a MWA versus RFA with better OS (HR: 1.55; 95% CI: 1.09–2.21, *p* = 0.01) and a better 2- and 3-year OS rate, 24-month PFS rate (Risk ratio [RR]: 0.67; 95% CI: 0.46–0.96, *p* = 0.03), and complete response rate (RR: 0.87; 95% CI: 0.79–0.96, *p* = 0.003) [115].

### 5.2. TACE and Portal Vein Embolization (PVE)

The concept of additional TACE on PVE is to increase the future liver remnant hypertrophy rate and to avoid tumor progression during the waiting period until hepatectomy. Several retrospective studies have compared the long-term outcome of TACE followed by PVE versus PVE alone followed by major hepatectomy and reported a significant advantage of a combined treatment with 5-year OS and recurrence free survival of 43–83.4% versus 31–57.7% and 37–61% versus 19–38%, respectively [116,117,118]. A recent intent-to-treat analysis investigating sequential TACE plus PVE versus PVE alone for patients with large HCC (≥5 cm) described a lower number of dropout patients for liver resection in the combination group 9% compared to 32% [119]. The OS was significantly better in the former compared to in the latter (3-year OS of 60% vs. 20%; *p* = 0.01).

### 5.3. TACE and Tyrosine-Kinase Inhibitors (TKI)

Tyrosine-kinase inhibitors are extensively used in the treatment of advanced HCC, and more precisely, in patients with vascular invasion and/or extrahepatic spread, it is well known that by inducing hypoxia leading to ischemic necrosis, chemoembolization activates the hypoxia-induced factors (HIFs) and increases the levels of the vascular endothelial factor (VEGF). As mentioned before, these angiogenic factors have been associated with poor prognosis, and they tend to increase more in suboptimal responders compared to in those with CR after TACE [54,55,56]. Accordingly, the hypothesis that combining TACE with antiangiogenics (including TKIs or antibodies to VEGF) was the basis of several trials. All of these successive studies, the “ POST-TACE” trial, conducted in Asia; the “SPACE” trial, a global venture; the British “TACE 2” trial; and the “STAH” trial from South Korea failed to show any benefits from adding sorafenib (multi-TKI, systemic first line treatment for advanced HCC) to TACE [120,121,122,123]. The SPACE and TACE 2 trials used the more standardized DEM-TACE procedure; nevertheless, no significant difference was seen in terms of time to progression (TTP) or PFS: 5.6 months vs. 5.5 months; HR 0.797 (95% CI 0.588–1.080); *p* = 0.072, and 7.8 months vs. 7.7 months; HR 1.03 (95% CI 0.75–1.42); *p* = 0.85, respectively [121,122].

The first and only study to support the combination of cTACE plus sorafenib is the recently published TACTICS trial, an open label, phase II, multicenter Japanese sRCT, whose co-primary endpoints were PFS and OS [124]. Nevertheless, PFS was not formulated as the standard definition but as time to untreatable (unTACEable) progression (TTUP); more precisely, it was defined as untreatable tumor progression, transient deterioration to Child–Pugh C, or the appearance of vascular invasion/extrahepatic spread. Considering this definition, the study described a significant difference in TTUP that was in favor of combination therapy: 25.2 months vs. 13.5 months; HR 0.59 (95% CI 0.41–0.87); *p* = 0.006. Nevertheless, this did not translate into better survival, as has been shown at ASCO 2021. However, the results of the TATICS trial exceed the data that has been so far described in the literature by far. One of the explanations given by the authors is the much longer median duration of sorafenib treatment (38.7 weeks compared to 17–21 months in the previously mentioned studies), owing to the TACE-specific trial design (sorafenib 400 mg/day 2–3 weeks prior to first TACE and discontinued for 2 days before and after each TACE session; dose augmentations to 800 mg/day were allowed after TACE according to tolerance). 

Other combinations with TKIs also failed to show a benefit in terms of PFS and TTP, such as brivatinib or orantinib [125,126]. 

It is safe to say that currently, there are no data to support the combined treatment of TACE and TKI therapy.

## 6. What’s Next for TACE?

Research in HCC is developing fast, and immune therapy has already proven its benefits for the treatment of advanced disease. Moreover, due to the astonishing results of the IMBRAVE 150 trial, a global, open-label, phase 3 trial that included 501 participants, the combination atezolizumab (anti-PDL1 checkpoint inhibitor) and bevacizumab (anti VEGF) is a new first line treatment for advanced HCC patients [127]. A recent update presented at ASCO GI 2021 showed a median OS of 19.2 months with the combination versus 13.4 months with sorafenib (HR, 0.66; 95% CI, 0.52–0.85; *p* = 0.0009). 

Could the association of TACE–immune therapy show an advantage compared to solo treatment? 

In 2007, Ayaru et al. were setting the basis of a possible partnership for TACE–immune therapy. His team showed that the necrosis produced by TACE unmasks tumor rejection Ag-specific T cell responses [128]. It generates an in situ immune response induction that could be combined with immunotherapy to increase the frequency of alpha-fetoprotein-specific T cells. This may lead to the control of tumor growth and survival improvement. Other data reported that strong tumor-associated antigen- (TAA) specific CD8+ T-cell responses generated by TACE therapy could suppress HCC recurrence. Thus, immune therapy to enhance TAA-specific CD8+ T-cells should be considered for clinical application in patients with HCC after local therapy [129]. Lastly, a new track could come from dendritic cell infusion after intra-arterial embolization [130]. Dendritic cell-based immunotherapies are believed to contribute to the eradication of the residual and recurrent tumor cells and have been proven to be safe for patients with cirrhosis and HCC and have been associated with lymphocyte and monocyte infiltration.

Several studies are underway that are analyzing different combinations of drugs and TACE, and their results are eagerly awaited (Table 3). IMMUTACE, a phase II single-arm, open-label study of TACE plus Nivolumab for BCLC B patients with preserved liver function, presented its first results at this year’s ESMO. The primary endpoint was OR according to mRECIST, with an OR > 55% (power = 80%; beta 0.17) being considered as promising for further investigation. *Vogel* et al. achieved an OR of 71.4% (95% CI), more precisely, 55.1% for CR and 16.3% of PR. These results open the path for combined treatment between TACE and immunotherapy. 

New research is also focusing on improving the TACE technique; thus, new biodegradable microspheres have been developed and are being tested. They are designed to preserve post-TACE target artery access, opening up the potential for cyclic treatment.

## 7. Future Is There One?

The evolution of TACE will probably not stop here, with multiple paths still being “under construction” and that are aiming at improving patient selection, new devices in order to standardize treatment, increasing selectivity in order to administer the chemotherapeutic agent more precisely and spare the non-tumoral liver, or combination treatments with newly developed immune therapy agents. The future of TACE will also be influenced by new developments in the field of selective internal radiation therapy (SIRT) since indications sometimes overlap. A more personalized treatment and the use of new isotopes (i.e., ^166^Holmium) are showing promising results and will increase the competition between these two intra-arterial treatments. Moreover, several studies are evaluating combination systemic treatments versus TACE (i.e., the German study, the ABC-HCC trial, NCT04803994, atezolizumab plus bevacizumab versus cTACE/DEM-TACE; the RENOTACE trial, regorafenib and nivolumab versus cTACE/DEM-TACE) and might threaten the future of chemoembolization. 

The results of current studies will guide the evolution of TACE in this era of systemic treatments with never-before-seen results and excellent tolerability. 

## Figures and Tables

**Figure 1 cancers-13-05129-f001:**
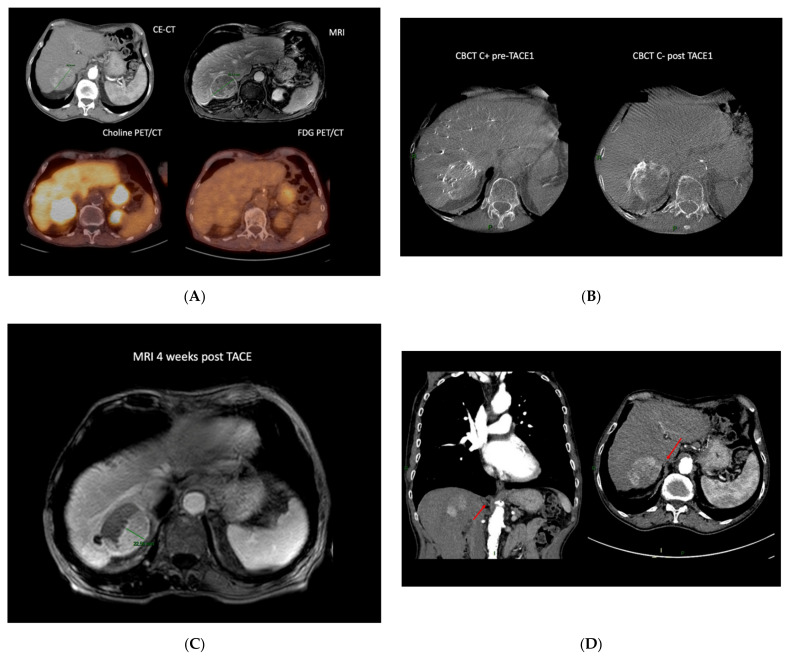
Case example of an 80-year-old patient diagnosed with a 62 mm HCC developed on cirrhosis (chronic hepatitis C). The work-up disclosed no distant metastasis. He underwent a double PET CT (FDG and choline) that showed a choline positive intake, suggesting a well-differentiated tumor (**A**). Due to multiple comorbidities, he could not undergo surgery; therefore, after a multidisciplinary discussion with the hepatic board, TACE was recommended. The enhanced CBCT performed before TACE showed sub-optimal treatment, with 30% of the lesion not being treated, which was confirmed on non-enhanced CBCT post DEM administration (**B**). The control MRI at 1 month showed a partial response with a 22.5 mm residual tumor (**C**). The patient had a CE-CT that showed a particular anatomy, with the non-responding side of the tumor being vascularized by a right phrenic artery (**D**). The patient underwent a second TACE, but unfortunately, the small artery could not be catheterized. Due to his general status, the patient benefited from BSC. (CE-CT: computed tomography; MRI: magnetic resonance imaging; Choline PET/CT: Choline positron emission tomography; FDG PET/CT: ^18^F-fluorodeoxyglucose positron emission tomography; CBCT C+: contrast enhanced cone-beam computed tomography; CBCT C-: non-enhanced cone-beam computed tomography; TACE: trans-arterial chemoembolization.)

**Table 3 cancers-13-05129-t003:** Summary of selected trials evaluating combination therapy of immune checkpoint inhibitors or/and tyrosin-kinase inhibitors and TACE.

Combination Therapy	Arms	Phase	Estimated Patient Number	Primary Endpoint	CilicalTrials.Gov Registration
Pembrolizumab plus TACE	NA	I/II	26	Incidence of adverse events	NCT03397654 (PETAL)
Durvalumab plus tremelimumab plus TACE	Durvalumab plus tremelimumab, durvalumab plus tremelimumab plus RFA/cryoablation/TACE	II	90	PFS	NCT02821754
Durvalumab plus tremelimumab plus DEM-TACE	Durvalumab plus tremelimumab plus DEM-TACE (two regimens)	II	30	OR	NCT03638141
Durvalumab plus bevacizumab plus TACE	Durvalumab plus bevacizumab plus TACE vs. TACE plus placebo	III	600	PFS	NCT03778957 (EMERALD-1)
Nivolumab plus DEM-TACE/TAE	Nivolumab plus DEM-TACE/TAE vs. DEM-TACE/TAE	III	522	OS	NCT04268888 (TACE-3)
Lenvatinib plus pembrolizumab plus cTACE	Lenvatinib plus pembrolizumab plus cTACE vs. cTACE	III	950	PFS-OS (co-primary)	NCT04246177 (LEAP-012)
Nivolumab plus Ipilimumab plus cTACE	Arm1 : nivolumab plus ipilimumab plus cTACE Arm2 : nivolumab plus placebo plus cTACE	III	765	TTTP-OS (co-primary)	NCT04340193 (CheckMate 74W)

cTACE: conventional transarterial chemoembolization; DEM-TACE: drug-eluting microspheres transarterial chemoembolization; RFA: radiofrequency ablation; OR: objective response; PFS: progression-free survival; OS: overall survival; TTTP: time to TACE progression or death.

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
