# Peer review of "In the Era of Systemic Therapy for Hepatocellular Carcinoma Is Transarterial Chemoembolization Still a Card to Play?"

_cancers, 2021, doi:10.3390/cancers13205129_

Round 1
Reviewer 1 Report
In this paper, the authors reviewed to explore the current data on the advancements of transarterial embolization (TACE) and the its future place amongst the growing panel of treatments.
This review may provide worthy scientific information on treatment of hepatocellular carcinoma.
I have nothing to additional comment.
Author Response
Thank you for your comments.
Reviewer 2 Report
the manuscript is very DEB-TACE heavy. The authors state throughout that DEB-TACE is safer, more tolerable and technically better but this is not supported by the data. This needs to be tempered.
-Remove acronyms in the title
-line 15 doesnt read correctly. please fix
line 26 - citation needed
line 34 citation needed
line 62 - this paragraph is repetitive with the one above. Correct either paragraph
3.1- heading doesnt make sense in context of paragraph
line 145 Child-Pugh is inconsistent throughout text - correct
3.3 a discussion of Art and HAP score is very necessary and clearly missing.
3.5 heading doesnt make sense. How are these techniques frenemies???
line 592 sorafenib is a generic so not capitals
TACTICs is the only study to illustrate survival benefit and I feel the reasons for this are very discussed in a very cursory fashion. Surely there are more reasons for success apart from more sorafenib?
no discussion of adjuvant immunotherapy trials, line 633 should refer to the NCT numbers for ongoing studies.
Whilst not a focus of the rv, the authors need to address SIRT as this is a key competitor in this space more than deb-tace
Author Response
the manuscript is very DEB-TACE heavy. The authors state throughout that DEB-TACE is safer, more tolerable and technically better but this is not supported by the data. This needs to be tempered.
Thank you for your comments. Here are the changes made according to your suggestions.
-Remove acronyms in the title
The acronyms have been removed.
-line 15 doesnt read correctly. please fix.
The phrase has been changed.
line 26 - citation needed
Citations for cTACE development are provided in the paragraph 3.1, more precisely reference 23 and 24.
line 34 citation needed
Citation is present in the dedicated paragraph.
line 62 - this paragraph is repetitive with the one above. Correct either paragraph
I feel the second paragraph is an extended explication of the first, including literature references.
3.1- heading doesnt make sense in context of paragraph
The heading refers to the fact that cTACE was the first technic to be described in the 1980s.
line 145 Child-Pugh is inconsistent throughout text – correct
It is true, it was changed to be homogenous.
3.3 a discussion of Art and HAP score is very necessary and clearly missing.
A paragraph has been added to discuss the STATE, HAP, ART and ABCR scores.
3.5 heading doesnt make sense. How are these techniques frenemies???
It seemed an interesting idea to present them as frenemies since they are variants of the same technic and literature is extensive on both, nevertheless debates are still ongoing on which is more effective or safer.
line 592 sorafenib is a generic so not capitals
This was changed.
TACTICs is the only study to illustrate survival benefit and I feel the reasons for this are very discussed in a very cursory fashion. Surely there are more reasons for success apart from more sorafenib?
According to the authors of this study, there were several reasons for the success of the combination group, as the particular design of the study and the modified PFS, as well as the fact that new lesions were not regarded as progressive disease. Furthermore, the administration of sorafenib before TACE had an effect on tumor vascularization, thus increasing the effects of intraarterial treatment. Nevertheless, this was not translated in a benefice in survival, according to the updates presented at ASCO 2021 (https://ascopubs.org/doi/abs/10.1200/JCO.2021.39.3_suppl.270).
no discussion of adjuvant immunotherapy trials, line 633 should refer to the NCT numbers for ongoing studies.
A table was added for the ongoing trials and we have presented to results of the IMMUTACE study that were revealed at this years ESMO.
Whilst not a focus of the rv, the authors need to address SIRT as this is a key competitor in this space more than deb-tace
The authors realize that SIRT is still a valid option in terms of intra-arterial locoregional treatment, nevertheless we avoided the debate TACE vs SIRT since this is a complex discussion and a full subject on its own. A phrase was added in the conclusion, for the perspectives of TACE (Paragraph 7).
Reviewer 3 Report
The review is well written and exhaustive. But I feel that something innovative is missing... I would only ask authors to remodulate the text of the 2 paragraphs on combined treatments: the text should be more detailed and it should give the readers precise informations, like timing of TACE and ablation (in the same session?Ablation followed by TACE, or TACE followed by ablation? after how many days?what kind of ablation technique should be used (RF, MW, Laser, etc)? and so on), similarly for TACE and systemic therapy (timing, drugs, etc).
Author Response
The review is well written and exhaustive. But I feel that something innovative is missing... I would only ask authors to remodulate the text of the 2 paragraphs on combined treatments: the text should be more detailed and it should give the readers precise informations, like timing of TACE and ablation (in the same session?Ablation followed by TACE, or TACE followed by ablation? after how many days?what kind of ablation technique should be used (RF, MW, Laser, etc)? and so on), similarly for TACE and systemic therapy (timing, drugs, etc).
Thank you for your comments. A Table was added in order to bring new information about combination treatments and ongoing studies.
Regarding the two paragraphs addressing combined treatments, the authors described that TACE should be performed before RFA, and added information the timing of ablation after chemoembolization, as found in the studies that offered this information. Nevertheless, the 2018 EASL guideline do not precise the delay between them.
The last paragraph of section 5.1 addresses the difference between TACE and RFA vs TACE vs MWA.
For systemic therapy, all the studies combining TACE and TKI were negative, that is why the authors did not detail the treatment program. Nevertheless, since the TACTICS trial is described in more detail, information about Sorafenib administration was added.
Round 2
Reviewer 3 Report
The text is more clear now